

# Evaluation of polarimetric ice microphysical retrievals with OLYMPEX campaign data

Armin Blanke[1], Andrew J. Heymsfield[2], Manuel Moser[3,4], and Silke Trömel[1,5]

[1]Institute of Geosciences, Department of Meteorology, University of Bonn, Bonn, 53121, Germany
[2]National Center for Atmospheric Research, Boulder, Colorado, USA
[3]Institute for Physics of the Atmosphere, University Mainz, Mainz, 55099, Germany
[4]Institut für Physik der Atmosphäre, Deutsches Zentrum für Luft- und Raumfahrt, Oberpfaffenhofen, 82234, Germany
[5]Laboratory for Clouds and Precipitation Exploration, Geoverbund ABC/J, Bonn, 53121, Germany

**Correspondence:** Armin Blanke (armin.blanke@uni-bonn.de)

**Abstract.** Polarimetric microphysical retrievals reveal a great potential for the evaluation of numerical models and data assimilation. However, the accuracy of ice microphysical retrievals is still poorly explored. To evaluate these retrievals and assess their accuracy, polarimetric radar measurements are spatially and temporally collocated with in situ aircraft measurements obtained during the OLYMPEX campaign (Olympic Mountain Experiment). Retrievals for ice water content IWC, total number

concentration $N_t$, and mean volume diameter $D_m$ of ice particles are assessed by comparing an in situ dataset obtained by the University of North Dakota (UND) Citation II aircraft with X-band Doppler on Wheels (DOW) measurements. Sector averaged range height indicator (RHI) scans are used to derive vertical profiles of microphysical retrievals. The comparison of these estimates with in situ data provides insights into strengths, weaknesses, and accuracy of the different retrievals, and quantifies the improvements of polarimetry-informed retrievals compared to conventional, non-polarimetric ones. In particular,

the recently introduced hybrid ice water content retrieval exploiting reflectivity $Z_{\mathrm{H}}$, differential reflectivity $Z_{\mathrm{DR}}$ and specific differential phase $K_{\mathrm{DP}}$ outperforms other retrievals based on either ($Z_{\mathrm{H}}$, $Z_{\mathrm{DR}}$) or ($Z_{\mathrm{H}}$, $K_{\mathrm{DP}}$) or non-polarimetric retrievals in terms of correlations with in situ measurements and the root mean square error.

## 1 Introduction

Polarimetric microphysical retrievals bear great potential for data assimilation and the evaluation of numerical models, how-

ever, their exploitation is still in its infancy. For instance, Trömel et al. (2021) demonstrated the potential of using polarimetric observations and retrievals to evaluate and improve microphysical parameterizations. Pioneering work by Carlin et al. (2016) revealed the benefits of assimilating polarimetric microphysical retrievals. Similar work is currently underway in Germany. Reimann et al. (2021) took a first step towards assimilating polarimetric variables into the ICOsahedral Nonhydrostatic (ICON) model (Zängl et al., 2015) via adapting microphysical retrievals for their application to observations of the polarimetric C-band

radar network of the German national meteorological service.

Ryzhkov et al. (1998) pointed to the limited database to identify the main reasons for the differences between individual in situ measurements and polarimetric retrievals. In fact, the in-depth evaluation of retrievals requires extensive airborne in situ



cloud particle measurements over polarimetric radar sites using, e.g., so-called Optical Array Probes (OAPs) collected during field campaigns. However, these are substantial high budget and provide data only along flight trajectories. Hogan et al. (2006)

introduced an ice water content (IWC) based on radar reflectivity Z and atmospheric temperature T. Tian et al. (2016), evaluated its performance along with that of a mean volume diameter $D_m$ retrieval with aircraft in situ data from the Bow Echo and Mesoscale Convective Vortex Experiment (BAMEX; Davis et al., 2004). They observed an overestimation of the mean Hogan IWC retrieval compared to in situ measurements (1.52 gm$^{-3}$ vs. 1.25 gm$^{-3}$) and a correlation of 0.55. Similarly the mean retrieved $D_m$ showed an overestimation (2.08 mm vs. 1.77 mm) and a low correlation of 0.27.

One reason for the bad performance of non-polarimetric retrievals is that the horizontal reflectivity $Z_{\mathrm{H}}$ in snow is approximately proportional to the fourth moment of the particle size distribution (PSD) (Hu and Ryzhkov, 2022), hence $Z_{\mathrm{H}}$ is insensitive to small particles, whereas other moments, such as the IWC, are sensitive to the small particle contributions. In contrast, specific differential phase $K_{\mathrm{DP}}$ which is proportional to the first moment of the PSD and thus the whole spectrum is considered. However, $K_{\mathrm{DP}}$ strongly depends on the aspect ratio and orientation of the particles, necessitating prior knowledge of these

parameters. Aydin and Tang (1997) proposed for IWC estimation the combination of $K_{\mathrm{DP}}$ and differential reflectivity $Z_{\mathrm{DR}}$, because their ratio is not affected by the variability of orientation and particles aspect ratio (Ryzhkov et al., 2018). Another set of polarimetric relations to quantify snow properties was derived by Bukovčić et al. (2018) exploiting $Z_{\mathrm{H}}$ and $K_{\mathrm{DP}}$. More recently, Carlin et al. (2021) suggested a hybrid application for estimating IWC by combining the complementary strengths and optimal ranges of the IWC retrievals following Bukovčić et al. (2018) and Ryzhkov and Zrnić (2019). Apart from IWC

retrievals, polarimetric retrieval relations have been suggested for $D_m$ and total number concentration of ice particles per unit volume $N_{\mathrm{t}}$ based on approaches utilizing combinations of three or two polarimetric variables, always including $K_{\mathrm{DP}}$ (Ryzhkov et al., 2018; Bukovčić et al., 2020). To evaluate the quality of polarimetric retrievals, approaches based on in situ and/or ground-based measurements were pursued.

Nguyen et al. (2019) proposed a methodology to retrieve IWC using $Z_{\mathrm{DR}}$ and $K_{\mathrm{DP}}$ from X-band dual-polarization airborne

radar data. This algorithm was found to be superior to power-law fits using Z against others, because $Z_{\mathrm{DR}}$ minimizes the dependence of IWC on variations in ice particle shape and orientation. Evaluation with in situ data from the High Altitude Ice Crystal – High Ice Water Content (HAIC-HIWC) field campaign revealed that the additional use of $Z_{\mathrm{DR}}$ reduced the root mean square difference by 6 % and the bias by 15 % on average compared to retrievals using $K_{\mathrm{DP}}$ only.

Several noise reducing techniques for reconstructing average vertical profiles from radar data have been proposed and used in

literature (Table 1). They are based on plan position indicator (PPI) or range height indicator (RHI) scans using single or multiple elevations. However, polarimetric and especially the phase-based radar measurements may be noisy in ice and snow and even more near the cloud top. As a consequence, it is beneficial to reduce their statistical noise before the calculation of microphysical retrievals. For instance columnar vertical profiles (CVPs; Murphy et al., 2020) represent local average vertical profiles that can be calculated at any distance from the radar using all elevation scans. Murphy et al. (2020) applied the microphysical

retrievals by Ryzhkov et al. (2018) to CVPs and tracked airborne in situ measurements to exploit them for evaluation. Overall, newly developed polarimetric retrievals show good promise in quantitatively estimating IWC, $N_{\mathrm{t}}$ and $D_m$. However, Murphy et al. (2020) especially revealed deficiencies near the melting layer (ML), resulting in, e.g., a pronounced underestimation of





**Table 1.** Overview of areal averaging techniques to derive quasi-vertical profiles of polarimetric radar variables.

| Methodology | Acronym | Scan strategy | Used azimuth | Used elevations | Citation |
|---|---|---|---|---|---|
| quasi-vertical profile | QVP | PPI | 360° | single high | Ryzhkov et al. (2016) |
| range-defined QVP | RD-QVP | PPI | 360° | multiple | Tobin and Kumjian (2017) |
| columnar vertical profile | CVP | PPI | sector | multiple | Murphy et al. (2020) |
| enhanced vertical profile | EVP | PPI | sector | multiple | Bukovčić et al. (2017) |
| slanted vertical profile | SVP | PPI | sector | single low | Bukovčić et al. (2017) |
| range-height-indicator-QVP | R-QVP | RHI | fixed | multiple | Allabakash et al. (2019) |

$D_m$ in these regions. Also using in situ measurements from aircraft, alternative retrieval methods were recently tested. Kedzuf et al. (2021) validated the accuracy of statistical polarimetric retrieval methods designed for pristine ice and aggregates and

Dunnavan et al. (2022) evaluated aspect ratio retrievals (Matrosov et al., 2020).

This study exploits measurements obtained during the Olympic Mountains Experiment (OLYMPEX) field campaign conducted from November 2015 to February 2016 on the Olympic Peninsula of Washington State, USA (Houze Jr et al., 2017). During OLYMPEX, the science aircraft University of North Dakota's (UND) Cessna Citation II equipped with an advanced in situ cloud payload performed overpasses over the National Science Foundation (NSF) funded Doppler On Wheels (DOW) radar.

We focus on the evaluation of radar-derived IWC, $D_m$ and $N_t$. Our accuracy assessments are based on two key aspects: 1) the matching of ground-based polarimetric radar data with airborne in situ cloud particle measurements in time and space, and 2) a noise-reducing averaging of the polarimetric radar measurements in the ice phase. Throughout OLYMPEX, the DOW X-band radar performed sequences of RHI scans in azimuthal sectors of 22°, which motivated us to introduce the RHI sector vertical profile (RSVP) technique to determine vertical profiles of polarimetric variables within specified sectors in azimuth and range.

The major objective of this study is to exploit the OLYMPEX campaign data for the accuracy assessment of the most recent microphysical retrievals and emphasize at the same time the benefits of X-band radars for microphysical studies.

The paper is organized as follows. Sect. 2 introduces the polarimetric remote sensing observations and airborne in situ measurements used. Sect. 3 summarizes the microphysical retrievals considered, while the RSVP methodology and the matching with aircraft measurements is detailed in Sect. 4. Evaluation results are presented in Sect. 5 followed by a discrepancy analyses

in Sect. 6. A summary of key findings and a comprehensive discussion of all results are provided in Sect. 7.

## 2   The OLYMPEX campaign data base

The OLYMPEX ground validation field campaign (Houze Jr et al., 2017), conducted in the Pacific Northwest, aimed at validating rain and snow measurements in midlatitude frontal systems and to further develop the Global Precipitation Measurement (GPM) mission satellite algorithms for precipitation estimation. Of the broad variety of ground instruments, including sev-





eral radars, and airborne instruments, the DOW in conjunction with the in situ aircraft measurements of the Citation II are appropriate for our evaluation purposes.

## 2.1 Polarimetric data from X-band radar DOW

The DOW radar (Wurman et al., 1997) placed within the Chehalis Valley at Lake Quinault (47.48° N, 123.86° W and at 64 m altitude), Washington, operated by the Center of Severe Weather Research (CSWR), is installed on a mobile truck.

The polarimetric dual-frequency X-band radar operated with a range of 59.96 km, a radial resolution of 75 m and used two independent transmitters at frequencies of $\sim$ 9.55 GHz and $\sim$ 9.40 GHz (Houze et al., 2018). In this study, we exclusively utilize the latter frequency as only measurements of the lower frequency were available after 12 November 2015. More detailed information on the DOW radar can be found in Houze et al. (2018).

The DOW's 10-minute scanning schedule includes plan position indicator (PPI) scans for the azimuthal sector between 39.2°

and 83.6° at six elevations between 2.8° to 11° and a series of 22 range-height indicators (RHIs) for the azimuthal sector between 50.4° and 71.4° in equidistant intervals of 1 degree and elevations ranging from 0° to 71°. For the evaluation of polarimetric ice microphysical retrievals, we focus in this study on the sector RHIs. All datasets utilized (Petersen et al., 2018), including a new version of the DOW data with improved calibration (Houze et al., 2018; doi: http://dx.doi.org/10.5067/ GPMGV/OLYMPEX/DOW/DATA201), were downloaded from the Global Hydrology Resource Center (GHRC) Distributed

Active Archive Center (DAAC). As a cross-check, we followed Ryzhkov and Zrnić (2019) and verified whether the expected $Z_\mathrm{H}$-$Z_\mathrm{DR}$ relationship for X-band in rain is fullfilled and made adjustments of $Z_\mathrm{DR}$ if necessary. Accordingly, a $Z_\mathrm{DR}$ correction was applied to all data used, with a larger correction required for measurements in November than in December (on average 0.38 dB vs. 0.19 dB).

Specific differential phase $K_\mathrm{DP}$, defined as half the range derivative of differential propagation phase shift $\Phi_\mathrm{DP}$, is estimated

following Vulpiani et al. (2012). For processing efficiency, the derivative of $\Phi_\mathrm{DP}$ is approximated using low-noise Lanczos differentiators (Holoborodko, 2008). Since $K_\mathrm{DP}$ is inversely proportional to the radar wavelength more reliable estimates can be expected at X-band compared to C- or S-band with according benefits also for microphysical retrievals. Only data with a cross-correlation coefficient $\rho_\mathrm{hv}$ above 0.7 are used for the $K_\mathrm{DP}$ estimation in order to reduce the impact of noisy and non-meteorological contamination.

## 2.2 In situ sensors and observations

The in situ microphysical cloud measurements during the OLYMPEX campaign have been acquired with the University of North Dakota (UND) Cessna Citation II airplane, which was equipped with an enhanced instrumental payload, including the 2D Stereo Imaging Probe (2D-S, SPEC Inc, USA) and the High Volume Precipitation Spectrometer (HVPS, SPEC Inc, USA). Both instruments provide shadowgraphs of cloud particles. Two-dimensional shadow images of hydrometeors are generated

as the particles penetrate through the sampling area of the particle imagers. A detailed description of the 2D-S and HVPS operating principles, uncertainties and limitations can be found in Lawson et al. (2006) and Baumgardner et al. (2017). Note that the data recorded by the 2D-S and HVPS differ in sampling volume size and pixel resolution. The 2D-S is equipped with



128 pixels of 10 μm resolution each, which allows imaging particles from 10 μm to 1.28 mm in size. With the larger pixel
resolution provided by the HVPS of 150 μm, imaging of hydrometeors from 150 μm to 3.25cm is enabled. Here, the dataset

of the horizontally oriented HVPS and the horizontally oriented part of the 2D-S processed by Heymsfield et al. (2018) were
combined with a switch-over size at 1000 μm, i.e. particles smaller 1000 μm are sized by the 2D-S and larger particles by
the HVPS. For a better comparability of the in situ measurements with the according radar-based retrievals of IWC, $D_m$ and
$N_t$, only particles larger than 100 μm are considered, because of the sensitivity of weather surveillance radars (Ryzhkov et al.,
2020) and uncertainties in probe data (Poellot and Bansemer, 2017; Baumgardner et al., 2017). With the particle size distribu-

tion (PSD) given by Heymsfield et al. (2018), $D_m$, IWC and $N_t$ for particles between 100 μm and 3 cm are calculated. In order
to derive $N_t$, the number concentration for each particle size bin is added up.

The different relations between measures of particle sizes used in the radar community and those derived from in situ mea-
surements must be considered in their comparison. Radar-based retrievals mostly provide the mean volume diameter $D_m$ as a
parameter for the size of the particles, which is defined as the ratio of the fourth to the third moment of the PSD:

$$D_m = \frac{\int D^4 N(D) dD}{\int D^3 N(D) dD}, \tag{1}$$

where $D$ is the equivolume diameter. It should be noted that $D_m$ is very close to the median volume diameter $D_0$. Assuming
that the PSD follows a gamma distribution with shape parameter $\mu$ (Ulbrich, 1983), we obtain

$$D_m = \frac{4+\mu}{3.67+\mu} D_0. \tag{2}$$

Instead of $D_m$, the aircraft microphysical probes usually measure either median mass diameter $D_{mm}$ or median volume diam-

eter $D_{mv}$ of the distribution of maximal particle dimension (Hu and Ryzhkov, 2022), which is given by

$$D_{max} = D\varphi^{-1/3}, \tag{3}$$

where $\varphi$ is the particle aspect ratio. For a gamma size distribution and taking Eq. (3) into account, we yield:

$$D_{mm} = \frac{2.67+\mu}{4+\mu} \frac{D_m}{\varphi^{1/3}}, \tag{4a}$$

$$D_{mv} = \frac{3.67+\mu}{4+\mu} \frac{D_m}{\varphi^{1/3}}, \tag{4b}$$

with Eq. (4a) assuming particle density being inversely proportional to its size. Assuming $\varphi = 0.6$ and an exponential distribu-
tion ($\mu = 0$), the following two $D_m$ relationships (Hu and Ryzhkov, 2022) are obtained:

$$D_m \approx \frac{D_{mm}}{0.79}, \text{ and} \tag{5a}$$

$$D_m \approx \frac{D_{mv}}{1.09}. \tag{5b}$$

These relations enable the direct comparison between the in situ measurements with the radar-based $D_m$ retrievals, with the

in situ derived median mass size $D_{mm}$ utilized in our study to obtain $D_m$. The IWC is estimated by using a mass-dimension
relation between mass $m$ and size given by

$$m = aD_{max}^b \tag{6}$$



with $a = 0.0121\,\text{kg/m}^b$, $b = 1.9$ and $\text{D}_{max}$ the diameter of the minimum enclosing circle of the projected 2D image. We followed Chase et al. (2018) by adopting Equation (6) and parameters $a$ and $b$ from Brown and Francis (1995) and modifying

them considering the particle size definition by Hogan et al. (2012).

In addition to the quantitative information provided by the 2D-S and HVPS, high-resolution in situ particle imagery data from the Cloud Particle Imager (CPI; SPEC Inc, USA) can be used to accurately identify particle types and characteristics. Only with the CPI it is possible to directly monitor supercooled liquid water (SLW) droplets of micron-size attached to ice particles and thus diagnose riming unambiguously and estimate the degree of riming.

The Rosemount Icing Detector (RICE; Baumgardner and Rodi, 1989) mounted on the Citation II is used as a supporting probe to detect the presence of SLW mandatory for riming (Vogel and Fabry, 2018). The RICE oscillates at a constant frequency, but when supercooled droplets freeze on its surface, the frequency of vibration decreases. Once accumulated ice exceeds a certain threshold, the probe tip is briefly heated to remove accreted ice. Data is available again as soon as the probe temperature has stabilized (Heymsfield and Miloshevich, 1989).

## 3   Radar-based microphysical retrievals

This section summarizes the most recent polarimetric and a suite of conventional non-polarimetric ice microphysical retrievals for IWC, $N_t$ and $D_m$ considered and assessed in this study. Two conventional $D_m$ retrievals derived from statistical relations between particle sizes and reflectivity expressed in linear scale ($Z_h = 10^{0.1Z_H}$; in units of $\text{mm}^6\text{m}^{-3}$) are used in our analysis. The first relation introduced by Skofronick-Jackson et al. (2019) is based on a power-law between $D_m$ (mm) and Ku-band

$Z_h$ fitted to data from the GPM Cold Season Precipitation Experiment (GCPEx) campaign (Skofronick-Jackson et al., 2015) conducted in Canada:

$$D_m{}^{\text{I}}(Z_h) = 1.45 Z_h{}^{0.25}. \tag{7}$$

Matrosov et al. (2019) introduced another power-law relation derived from $Z_h$ data of ground-based S-band radar and aircraft in situ calculated $D_{mv}$ obtained during the Indirect and Semi-Direct Aerosol Campaign (ISDAC) in Alaska (Maahn et al.,

2015). Assuming $\varphi = 0.6$ and using Eq. (5b), it follows

$$D_m{}^{\text{II}}(Z_h) = \frac{1}{1.09} \cdot \left(1.15 Z_h{}^{0.271}\right). \tag{8}$$

This equation differs from Eq. (5) in Murphy et al. (2020), likely due to confusion among experts regarding the conversion of $D_{mm}$ to $D_m$.

Since $Z_H$-based $D_m$ retrievals are disproportionally weighted by a few large particles, polarimetric $K_{DP}$-based retrievals have a

great potential to provide more accurate estimates. Additionally, such estimators use the key advantage that $K_{DP}$ is not affected by attenuation and not biased by noise and radar miscalibration. To retrieve a polarimetric $D_m$ from $K_{DP}$ and $Z_{dp}$, where $Z_{dp} = Z_h$ - $Z_v$ is the reflectivity difference at horizontal and vertical polarization in linear scale, as proposed in Ryzhkov et al. (2018)





and Ryzhkov and Zrnić (2019) we use

$$D_m(Z_{\mathrm{dp}}, K_{\mathrm{DP}}) = -0.1 + 2.0 \left( \frac{Z_{\mathrm{dp}}}{K_{\mathrm{DP}}\lambda} \right)^{1/2}, \tag{9}$$

where $K_{\mathrm{DP}}$ is in $°\mathrm{km}^{-1}$, $Z_{\mathrm{dp}}$ is in $\mathrm{mm}^6\mathrm{m}^{-3}$ and the radar wavelength $\lambda$ in mm. This estimator is largely immune to variations in ice particle orientation and shape but has the inherent deficiency of being impacted by the degree of riming, therefore it is supposed to be more appropriate for lower temperature regions where riming is less likely. As an alternative, Bukovčić et al. (2018, 2020) uses $Z_{\mathrm{h}}$ and $K_{\mathrm{DP}}$ to retrieve

$$D_m(Z_{\mathrm{h}}, K_{\mathrm{DP}}) = 0.67 \left( \frac{Z_{\mathrm{h}}}{K_{\mathrm{DP}}\lambda} \right)^{1/3}. \tag{10}$$

Unlike $D_m(Z_{\mathrm{dp}}, K_{\mathrm{DP}})$, however, this retrieval is not immune to the variability in particle orientation and shape, a strength of this $D_m(Z_{\mathrm{h}}, K_{\mathrm{DP}})$ estimate is that it does not depend on density and is therefore not affected by the degree of riming.

Similarly to the aforementioned $D_m$ retrievals, IWC retrievals can also be derived in a purely empirical fashion, through utilization of power-laws. Hogan et al. (2006) introduced an expression which related in situ measured IWC (in g m$^{-3}$) to reflectivity (in dBZ) at various frequencies (e.g., 3 GHz) and temperature $T$ (in °C) in the European Cloud Radiation

Experiment (EUCREX) derived in the Rayleigh approximation:

$$\log_{10}\left(\mathrm{IWC}^{\mathrm{I}}(Z_{\mathrm{H}}, T)\right) = 0.06Z_{\mathrm{H}} - 0.0197T - 1.7. \tag{11}$$

They also exploited the $\mathrm{IWC}^{\mathrm{I}}(Z_{\mathrm{H}}, T)$ to evaluate the mesoscale version of the Met Office Unified Model. However, the IWC relationship implicit in the models parameterization (see Hogan et al. (2006) for details) is

$$\log_{10}\left(\mathrm{IWC}^{\mathrm{II}}(Z_{\mathrm{H}}, T)\right) = 0.06Z_{\mathrm{H}} - 0.0212T - 1.92. \tag{12}$$

Nguyen et al. (2019) proposed two more empirical, but polarimetric IWC retrievals via optimal fitting parameters, exploiting aircraft measurements from a polarimetric side pointing X-band radar and measured IWC by in situ probes obtained during HAIC-HIWC. The two retrievals utilize either $K_{\mathrm{DP}}$ only or include additionally the differential reflectivity $Z_{\mathrm{dr}}$ expressed in linear scale ($Z_{\mathrm{dr}} = 10^{0.1 Z_{\mathrm{DR}}}$):

$$\mathrm{IWC}(K_{\mathrm{DP}}) = 0.903 K_{\mathrm{DP}} + 0.319, \text{ and} \tag{13}$$

$$\mathrm{IWC}^{\mathrm{I}}(Z_{\mathrm{dr}}, K_{\mathrm{DP}}) = \frac{0.136 K_{\mathrm{DP}} + 0.037}{1 - Z_{\mathrm{dr}}^{-1}}. \tag{14}$$

$Z_{\mathrm{dr}}$ is set to 1.15 (Ryzhkov et al., 1998) when $Z_{\mathrm{dr}}$ falls below this threshold. Ryzhkov et al. (1998) also demonstrated that $K_{\mathrm{DP}}$ is sensitive to the aspect ratio $\varphi$ and orientation of the particles, whereas IWC is not, requiring additional knowledge about the particles. Accordingly, the estimator $\mathrm{IWC}(K_{\mathrm{DP}})$ is highly affected by variations in $\varphi$ and/or orientation, whereas the inclusion of $Z_{\mathrm{dr}}$ in $\mathrm{IWC}^{\mathrm{I}}(Z_{\mathrm{dr}}, K_{\mathrm{DP}})$ reduces these dependences. The observational study by Nguyen et al. (2019) demonstrated that their

empirical relation $\mathrm{IWC}^{\mathrm{I}}(Z_{\mathrm{dr}}, K_{\mathrm{DP}})$ is very close to the theoretical IWC relation by Ryzhkov et al. (2018):

$$\mathrm{IWC}^{\mathrm{II}}(Z_{\mathrm{dr}}, K_{\mathrm{DP}}) = 4 \times 10^{-3} \left( \frac{K_{\mathrm{DP}}\lambda}{1 - Z_{\mathrm{dr}}^{-1}} \right). \tag{15}$$



The latter exploits the inherent information of $Z_{dr}$ about shape and orientation of particles. Similar to Eq. (9), $\text{IWC}^{II}(Z_{dr}, K_{DP})$ is practically insensitive to the shape and orientations of the ice particles, because the numerator and denominator are proportionally impacted and thus the ratio is not affected.

In absence of a birdbath scan in the scan schedule (as it is e.g. the case for the operational radar networks in the United States), ensuring high $Z_{dr}$ accuracy is often difficult. Bukovčić et al. (2018, 2020) introduced a generalized relation for IWC

$$\text{IWC}(Z_h, K_{DP}) = \frac{10.2 \times 10^{-3}}{(F_0 F_S)^{0.66}} (K_{DP}\lambda)^{0.66} Z_h^{0.28}, \tag{16}$$

which is immune to $Z_{dr}$ miscalibrations and where $F_0$ is the orientation factor as function of the width of the canting angle distribution $\sigma$ and $F_S$ the shape factor determined by $\varphi$. The relation in (16) produces

$$\text{IWC}(Z_h, K_{DP}) \approx 0.31 K_{DP}^{0.66} Z_h^{0.28}, \tag{17}$$

for $\sigma = 0°$, $\varphi = 0.65$, and $\lambda = 32$ mm. In contrast to Eq. (10), however, the $\text{IWC}(Z_h, K_{DP})$ retrieval is sensitive to the density of the particles and thus, the degree of riming.

The combined application of Eq.(15) in regions where $Z_{DR}$>0.4 dB and Eq.(17) elsewhere, as suggested by Carlin et al. (2021), leverages the strengths of both formulas and denoted as $\text{IWC}_{\text{Carlin}}$ estimator in the following.

Estimating snow concentration $N_t$ is a challenging task and it is almost impossible to derive it with acceptable accuracy from single-polarization radar measurements, because of the wide variety of ice and snow habits and their microphysical properties. A polarimetric retrieval for the logarithm of the total number concentration of ice particles $N_t$ (in $L^{-1}$) following Ryzhkov et al. (2018, 2019) is estimated by

$$\log_{10}\left(N_t(Z_H, Z_{dp}, K_{DP})\right) = 0.1Z_H - 2\log_{10}\gamma - 1.33 \quad \text{with} \tag{18a}$$

$$\gamma \approx 0.78\left(\frac{Z_{dp}}{K_{DP}\lambda}\right). \tag{18b}$$

Again, using the ratio of $Z_{dp}/K_{DP}$ in this retrieval cancels out the effects of orientation and shape. Another relation for $N_t$ included in our accuracy assessment is given by Carlin et al. (2021):

$$\log_{10}\left(N_t(Z_H, \text{IWC})\right) = 6.69 + 2\log_{10}(\text{IWC}) - 0.1Z_H. \tag{19}$$

Combined with $\text{IWC}_{\text{Carlin}}$, Eq. (19) is hereafter referred to as $N_t(Z_H, \text{IWC}_{\text{Carlin}})$. Note that polarimetric retrieval equations (Eq.(9), Eq.(10) and Eq.(15)–(18a)) were derived in the Rayleigh approximation assuming that the density $\rho_s$ of Rayleigh scatterers (ice particles, snowflakes) is inversely proportional to D, according formula following Brandes et al. (2007):

$$\rho_s(D) = \alpha_0 f_{\text{rim}} D^{-1}, \tag{20}$$

where $\rho_s$ is expressed in g cm$^{-3}$ and the factor $\alpha_0$ varies with the degree of riming $f_{\text{rim}}$, which ranges from 1 for unrimed ice to 5 for heavily rimed ice. For larger non-Rayleigh scatterers like graupel or hail, these polarimetric retrieval equations are not valid (Ryzhkov et al., 2020). Reliable retrievals are only obtainable in areas where $Z_{DR}$ and $K_{DP}$ are not very close to zero,



**Table 2.** Table of retrieval equations used in this work. $D_m$ is in units of mm, IWC is in units of $\mathrm{g\,m^{-3}}$ and $N_t$ is in units of $\mathrm{L^{-1}}$; $T$ is in units of °C.

| Retrieval | Formula | Type | Reference |
|---|---|---|---|
| $D_m^{\mathrm{I}}(Z_h)$ | $1.45 Z_h^{\,0.25}$ | empirical | Skofronick-Jackson et al. (2019) |
| $D_m^{\mathrm{II}}(Z_h)$ | $1.06 Z_h^{\,0.271}$ | empirical | Matrosov et al. (2019) |
| $D_m(Z_{dp}, K_{DP})$ | $-0.1 + 2.0 \left( \frac{Z_{dp}}{K_{DP}\lambda} \right)^{1/2}$ | empirical | Ryzhkov and Zrnić (2019) |
| $D_m(Z_h, K_{DP})$ | $0.67 \left( \frac{Z_h}{K_{DP}\lambda} \right)^{1/3}$ | empirical | Bukovčić et al. (2020) |
| $\log_{10}(\mathrm{IWC^I}(Z_H, T))$ | $0.06 Z_H - 0.0197 T - 1.7$ | empirical | Hogan et al. (2006) |
| $\log_{10}(\mathrm{IWC^{II}}(Z_H, T))$ | $0.06 Z_H - 0.0212 T - 1.92$ | model | Hogan et al. (2006) |
| $\mathrm{IWC}(K_{DP})$ | $0.903 K_{DP} + 0.319$ | empirical | Nguyen et al. (2019) |
| $\mathrm{IWC^I}(Z_{dr}, K_{DP})$ | $(0.136 K_{DP} + 0.037)\left(\frac{1}{1 - Z_{dr}^{-1}}\right)$ | empirical | Nguyen et al. (2019) |
| $\mathrm{IWC^{II}}(Z_{dr}, K_{DP})$ | $4 \times 10^{-3} \left( \frac{K_{DP}\lambda}{1 - Z_{dr}^{-1}} \right)$ | theoretical | Ryzhkov and Zrnić (2019) |
| $\mathrm{IWC}(Z_h, K_{DP})$ | $0.31 K_{DP}^{0.66} Z_h^{\,0.28}$ | empirical | Bukovčić et al. (2020) |
| $\mathrm{IWC_{Carlin}}$ | $\begin{cases} \mathrm{IWC^{II}}(Z_{dr}, K_{DP}) \text{ if } Z_{DR} > 0.4 \text{ dB}, \\ \mathrm{IWC}(Z_h, K_{DP}) \text{ otherwise} \end{cases}$ | theoretical and empirical | Carlin et al. (2021) |
| $\log_{10}(N_t(Z_H, Z_{dp}, K_{DP}))$ | $0.1 Z_H - 2\log_{10}\gamma - 1.33$ | theoretical | Ryzhkov and Zrnić (2019) |
| $\log_{10}(N_t(Z_H, \mathrm{IWC}))$ | $6.69 + 2\log_{10}(\mathrm{IWC}) - 0.1 Z_H$ | theoretical and empirical | Carlin et al. (2021) |

which represents a weakness of polarimetric retrievals. In addition, a recent study showed that polarimetric ice microphysical retrievals following Ryzhkov and Zrnić (2019) provide best results at cold temperatures, i.e. lower than -10°C to -15°C (Murphy et al., 2020). Around this temperature interval the dendritic growth layer (DGL) is located and $Z_{DR}$ and $K_{DP}$ exhibit pronounced signals. For this reason, our analysis is restricted to temperatures below -10°C.

## 4  Methodology


RHI scans obtain vertically high-resolved measurements and thus are well suited for microphysical studies and our accuracy assessment. Furthermore, the sequences of successive RHIs performed during the OLYMPEX campaign in azimuthal sectors of 22 degrees provide high-resolution 3D measurements in this predefined region. Our newly introduced RHI sector vertical profile (RSVP) technique provides noise-reduced quasi-vertical profiles of polarimetric variables obtained by azimuthal aver-

aging of RHI sector scans in a convenient height versus time format. This method was inspired by both, the QVP methodology and the ability of CVPs to follow flight segments of research aircraft, and on top takes advantage of the RHI scan mode. Figure 1 illustrates the RSVP technique introduced here for the matching with airborne in situ measurements.

To create a RSVP, a series of RHIs is first averaged within the azimuthal range $\alpha$. The range of elevation angles $\theta$ considered can be adapted, e.g. to minimize ground clutter effects.

In the second step, windows of the desired size and position are selected along the range axis of the already azimuthally aver-





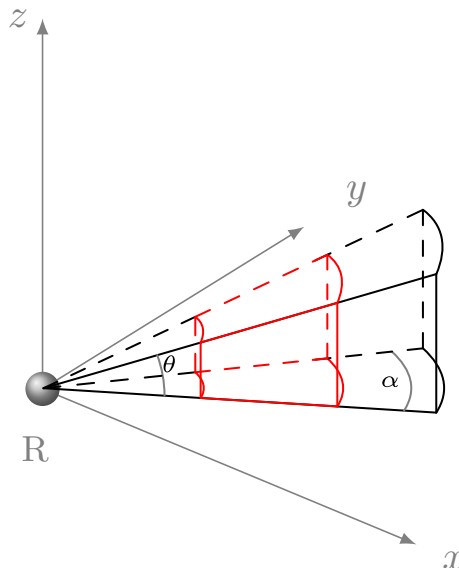

**Figure 1.** RSVP column (outline in red), covering an arbitrary volume in range in azimuth.

aged RHI (marked in red in Fig. 1), and averaged along the chosen range interval as well. Based on the selected and averaged volume, a mean profile is computed representing the vertical columns that are combined and displayed in the RSVP. In this way, RSVPs enable both the joint analysis with fixed ground-based measurements provided by vertically pointing devices like e.g., Micro Rain Radars (MRRs), and the tracking of research aircraft within the sector covered by the RHIs for the matching

with airborne measurements. In the latter case, the selected volume taken into account in the averaging process changes with time. Additionally, vertical averaging is applied, with 75 m bins to match the aircraft track and account for aircraft altitude fluctuations. Four columns with a length on the range axis of 4.5 km each, starting at 2 km distance of the radar are used for tracking the aircraft. The first 2 km were omitted, because of known inconsistencies in the transmitters and reduced polarimetric information content. Also, the maximum range considered in this analysis was 20 km from the radar in order to

reduce partial beam blockage by surrounding mountains. The temporal resolution of the RSVP technique depends for sure on the scan schedule. During the OLYMPEX campaign, the 22 RHI measured in the azimuthal sector were available every 4 min interspersed with a 2 min PPI scan after each two RHI sector scans.

Figure 2 shows as an example RSVPs of $Z_H$, $Z_{DR}$, $\rho_{hv}$ and $K_{DP}$ for a complex occluded front observed on 18 December 2015. Similar RSVPs have been generated for all 20 flights during OLYMPEX totaling approximately 60 flight hours (not shown

here). The event displayed in Fig. 2 exhibits throughout clearly visible ML signatures in $Z_H$, $Z_{DR}$ and $\rho_{HV}$, roughly following the temporal evolution of the 0°C isotherm. Within the DGL, located at temperatures between -10°C and -15°C, also bands of enhanced $Z_{DR}$ and $K_{DP}$ are visible.

For the accuracy assessment, the microphysical retrievals introduced in Sect. 3 are calculated based on the RSVPs and dis-



played in a similar manner, considering only data above the ML. For this purpose, only radar data with $Z_{DR}> 0.1$ dB, $Z_H> 0$

dBZ, $K_{DP} > 0.01°km^{-1}$ and $\rho_{hv}> 0.7$ is used.

A direct comparison of airborne in situ measurements with ground-based retrievals requires a careful matching in both space and time. Research aircraft measure along flight trajectories, while the RSVP technique uses stationary ground-based radars monitoring a volume at flight altitude. The aircraft measurements of IWC, $D_m$, and $N_t$ are averaged along the respective flight path sections and compared to the radar retrievals in the according columns and at the corresponding aircraft altitude. Similarly,

the flight altitude and measured environmental parameters (e.g., temperature) are averaged for the time intervals within each column. The temperature information enables us to exclude bright band effects. Herein, the averaged in situ observations are assumed to be characteristic of the entire collocated radar volume. Figure 3 illustrates the matching of microphysical retrievals based on RSVP data with airborne in situ measurements. During OLYMPEX, the UND Citation II research aircraft performed 148 transects over DOW at different altitudes. In our study, only the aircraft measurements between 3 km and 7 km height are

used. The use of RSVP columns allows analysis of multiple collocated data points within a single overpass. Here, the selected length of 4.5 km along the range for the RSVP columns ensures an aircraft flying at approximately 100 m s$^{-1}$ is sufficiently long within the column (t≥30 s). Flight intercepts of less than 30 s in duration were discarded from the analysis as they may

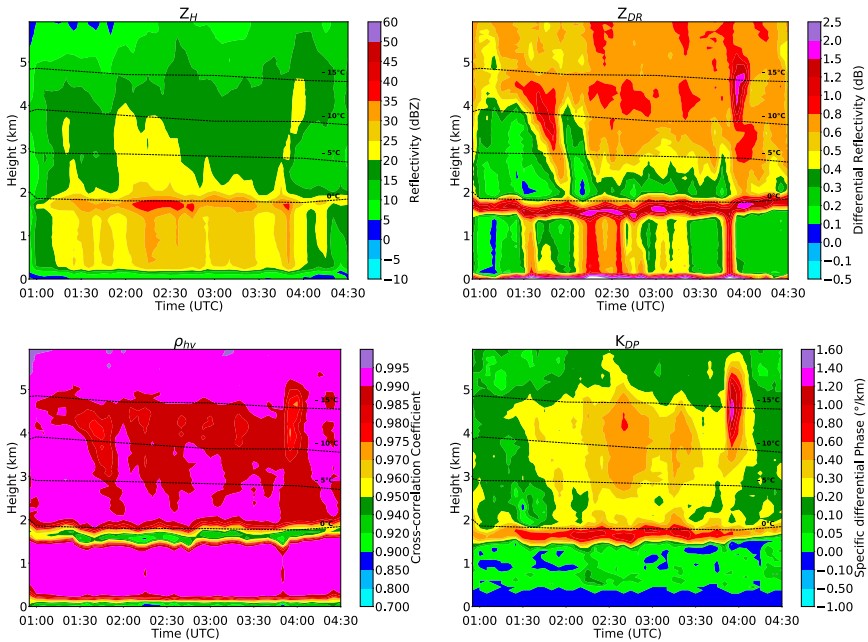

**Figure 2.** RSVPs of $Z_H$ (top left), $Z_{DR}$ (top right), $\rho_{hv}$ (bottom left), and $K_{DP}$ (bottom right) using measurements from the DOW on 18 December 2015 between 00:40 and 04:30 UTC. The time interval shown is equal to the flight mission length, to nearest 15 min. Data in the RSVPs are at ranges from 6.5 km to 11 km away from the radar and 22° in azimuth. The overlaid dashed lines (in all panels) display the 0°C, -5°C, -10°C and -15°C isotherms from European Centre for Medium-Range Weather Forecasts Reanalysis v5 (ERA5, Hersbach et al., 2020) at the DOW location.





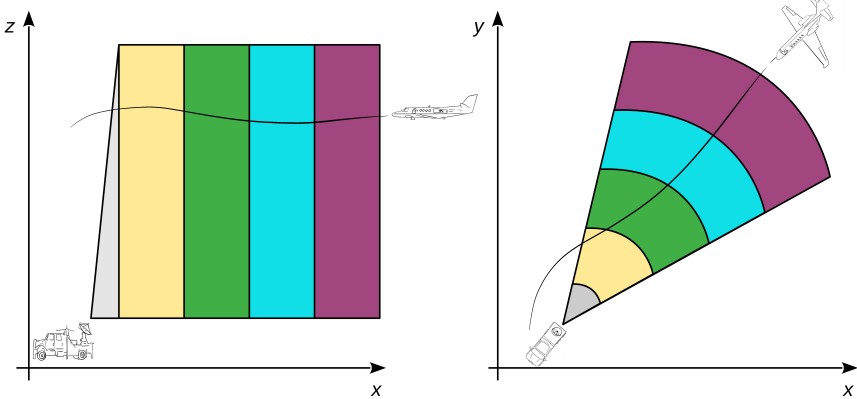

**Figure 3.** A schematic illustration from the collocation of RSVP columns with in situ measurements. Each colored column can be created according to the RSVP procedure depicted in Fig. 1. Gray shaded areas represent omitted data very close to the radar. The left panel shows the side view and the right panel the top view of the aircraft flying through the columns.

not adequately represent the respective RSVP columns and to ensure statistical reliability. Too short flight intercepts occurred, e.g., when the aircraft shortly flew along the edge of a column and then left the sector again (e.g., yellow column in right panel of Fig. 3) or did not pass the RHI sector at all. Note that limited temporal resolution of RSVPs (here 4 min) and comparatively short flight segments can lead to a temporal mismatch between the two datasets of up to 3 min in this study.

## 5 Accuracy assessment of the ice microphysical retrievals

This section presents the resulting accuracies for all polarimetric and non-polarimetric retrievals outlined in Sect. 3 exploiting the collocated in situ measurements available during the OLYMPEX campaign. The in situ measurements are assumed to be the truth in this study despite similar existing uncertainties, e.g., with the assumed mass–dimension relationship. As statistical measures for the agreement between the in situ measurements and the different radar-derived retrievals, the mean and median retrieved-to-measured ratio (RMR), the mean difference (bias), the root mean square error (RMSE), and Pearson's correlation coefficient $r$ are considered. Results for all ice microphysical retrievals introduced in Sect. 3 are shown in Table 3. The retrieval analysis identifies in terms of RMSE IWC$_{\mathrm{Carlin}}$, $N_{\mathrm{t}}(Z_{\mathrm{H}},\mathrm{IWC}_{\mathrm{Carlin}})$ and $D_m(Z_{\mathrm{dp}}, K_{\mathrm{DP}})$ as the best performing set for the three quantities considered. It can be seen that the use of polarimetry clearly improves the estimates of IWC and $D_m$ compared to the conventional, non-polarimetric retrievals. In particular, IWC$_{\mathrm{Carlin}}$ can improve $r$ by 7 % over IWC$^{\mathrm{I}}(Z_{\mathrm{H}},T)$ and reduces RMSE by 37 %. An even greater improvement occurs for the estimation of $D_m$, where the use of $D_m(Z_{\mathrm{dp}}, K_{\mathrm{DP}})$ over both non-polarimetric $D_m$ retrievals increases $r$ by 15 % compared to both non-polarimetric $D_m$ retrievals. $D_m(Z_{\mathrm{dp}}, K_{\mathrm{DP}})$ also brings a reduction of RMSE by 47 % compared to $D_m{}^{\mathrm{I}}(Z_{\mathrm{h}})$. Analysis of IWC$^{\mathrm{I}}(Z_{\mathrm{dr}}, K_{\mathrm{DP}})$ in the ice regions of tropical clouds





**Table 3.** Correlations ($r$), slopes and intercepts from least-squares fits, root-mean-square error (RMSE), biases, and mean and median retrieved-to-measured ratio (RMR) for each microphysical property and for all retrievals. Best values of each statistical measure across every microphysical retrieval type are highlighted in boldface font.

| IWC (gm$^{-3}$) | r | Slope | Intercept | RMSE | Bias | RMR Mean | RMR Median | Publication |
|---|---|---|---|---|---|---|---|---|
| IWC$^\mathrm{I}(Z_\mathrm{H}, T)$ | 0.90 | 0.88 | -0.04 | 0.30 | -0.11 | 1.23 | 1.59 | Hogan et al. (2006) |
| IWC$^\mathrm{II}(Z_\mathrm{H}, T)$ | 0.91 | 1.40 | -0.05 | 0.30 | 0.10 | 0.79 | **1.00** | Met Office Model |
| IWC$_\mathrm{Comb}(Z_\mathrm{H}, T)$ | 0.95 | 1.18 | -0.06 | 0.20 | **0.02** | **0.95** | 1.11 | IWC$^\mathrm{I}(Z_\mathrm{H}, T)$ & IWC$^\mathrm{II}(Z_\mathrm{H}, T)$ |
| IWC$(K_\mathrm{DP})$ | **0.98** | 1.29 | -0.42 | 0.28 | -0.22 | 1.46 | 2.02 | Nguyen et al. (2019) |
| IWC$^\mathrm{I}(Z_\mathrm{dr}, K_\mathrm{DP})$ | 0.97 | 1.20 | -0.34 | 0.26 | -0.21 | 1.43 | 1.97 | Nguyen et al. (2019) |
| IWC$^\mathrm{II}(Z_\mathrm{dr}, K_\mathrm{DP})$ | 0.94 | 0.87 | **-0.02** | 0.24 | -0.10 | 1.21 | 1.25 | Ryzhkov et al. (2018) |
| IWC$(Z_\mathrm{H}, K_\mathrm{DP})$ | 0.94 | **1.00** | -0.10 | 0.23 | -0.10 | 1.22 | 1.76 | Bukovčić et al. (2018) |
| IWC$_\mathrm{Carlin}$ | 0.96 | 1.03 | -0.05 | **0.19** | -0.04 | 1.08 | 1.13 | Carlin et al. (2021) |
| N$_t$ log$_{10}$(L$^{-1}$) | | | | | | | | |
| N$_t(Z_\mathrm{H}, Z_\mathrm{dp}, K_\mathrm{DP})$ | 0.88 | **1.13** | **-0.33** | 0.46 | -0.18 | 1.17 | 1.33 | Ryzhkov et al. (2018) |
| N$_t(Z_\mathrm{H}, \mathrm{IWC}_\mathrm{Carlin})$ | **0.91** | 1.38 | -0.52 | **0.43** | **-0.09** | **1.08** | **1.27** | Carlin et al. (2021) |
| D$_\mathrm{m}$ (mm) | | | | | | | | |
| D$_\mathrm{m}^\mathrm{I}(Z_\mathrm{h})$ | 0.79 | 0.55 | 0.04 | 2.12 | -1.82 | 1.78 | 1.86 | Skofronick-Jackson et al. (2019) |
| D$_\mathrm{m}^\mathrm{II}(Z_\mathrm{h})$ | 0.79 | **0.64** | 0.19 | 1.40 | -0.99 | 1.42 | 1.48 | Matrosov et al. (2019) |
| D$_\mathrm{m}(Z_\mathrm{h}, K_\mathrm{DP})$ | **0.94** | 2.60 | -0.85 | 1.38 | 1.10 | 0.53 | 0.54 | Bukovčić et al. (2018) |
| D$_\mathrm{m}(Z_\mathrm{dp}, K_\mathrm{DP})$ | 0.91 | 1.59 | **0.01** | **1.13** | **0.87** | **0.63** | **0.74** | Ryzhkov et al. (2018) |

during 7 flights of the HAIC-HIWC field campaign in Cayenne, French Guiana, showed an overall correlation between in situ and estimated IWCs of 0.72 and a mean RMSE of 0.52 gm$^{-3}$ (Nguyen et al., 2019). In our case, a lower RMSE of 0.26 gm$^{-3}$ can be observed for their IWC$^\mathrm{I}(Z_\mathrm{dr}, K_\mathrm{DP})$ and a systematically higher correlation of 0.97. Again, the advantages of using polarimetry over non-polarimetry are evident in this retrieval. It is interesting that IWC$(K_\mathrm{DP})$ shows the highest correlation with 0.98, although $Z_\mathrm{dr}$ is not included. Nevertheless, both polarimetric estimators based on optimal fitting parameters exhibit higher RMSE compared to all other polarimetric IWC retrievals, manifested in a systematic overestimation. A possible explanation for the overestimation may be, that they were optimized for the tropical climate region characterized by an average higher IWC.

For a more detailed look at the evaluation procedure to obtain information about quantitative statistics and accuracies, we consider an example (Fig. 4) for the best performing IWC retrieval. Out of 20 campaign flights, 10 flight missions were selected and listed in Table 4 that have good data quality (in situ and radar) and met all filter criteria. With the thresholds applied, the correlation of IWC$_\mathrm{Carlin}$ improves from r=0.91 to r=0.96. The slope of the filtered regression is 1.03, which is closer to the 1 to 1 line than the unfiltered green regression line. The important information here is that the filtering via thresholds provides us a





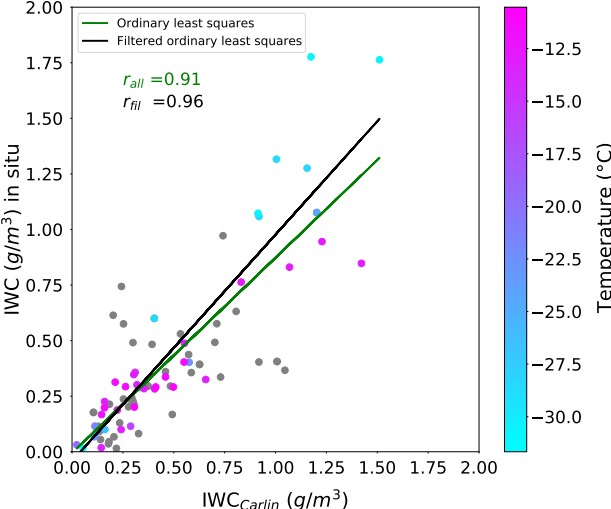

**Figure 4.** Scatter plots and linear regressions of IWC$_{Carlin}$ vs. IWC from OLYMPEX in situ data. Temperature-dependent colouring of each data point indicates temperatures colder than or equal to -10°C, while the gray dots, represent data masked via filter criteria for temperature (T≥-10°C) and intersection time (t≤30 s). The $r$ statistics are reported in green for all data points and in black for filtered ones.

**Table 4.** List of selected Citation II flight missions after applying filter criteria with resulting collocated data points.

| Dates | Flight times (UTC; to nearest 15 min) | data points |
|---|---|---|
| 12 November | 19:30-22:30 | 0-1 |
| 13 November | 15:00-17:45 | 2-3 |
| 01 December | 22:45-01:45 | 4-8 |
| 04 December | 13:00-16:00 | 9-11 |
| 05 December | 14:45-18:00 | 12-14 |
| 10 December | 14:45-17:00 | 15-22 |
| 12 December | 17:00-20:15 | 23-26 |
| 13 December | 15:45-19:15, 20:00-23:15 | 27-34 |
| 18 December | 01:15-04:30 | 35-37 |

more reliable database for the analysis.

The repeated analysis without temperature threshold applied showed a significant decrease in the correlations for all retrievals

(not shown). Figure 5 shows this best performing set of ice microphysical retrievals together with the in situ data. Overall, we see a tendency towards a slight overestimation of IWC at warmer and an underestimation at colder temperatures, with largest in situ standard deviation at colder temperatures. IWC$_{Carlin}$ yields a high correlation of r = 0.96, the lowest RMSE of 0.19 gm$^{-3}$ and a near-zero bias of -0.04 gm$^{-3}$. Carlin et al. (2021) found some evidence that using IWC$_{Carlin}$ for the initialization of a 1D





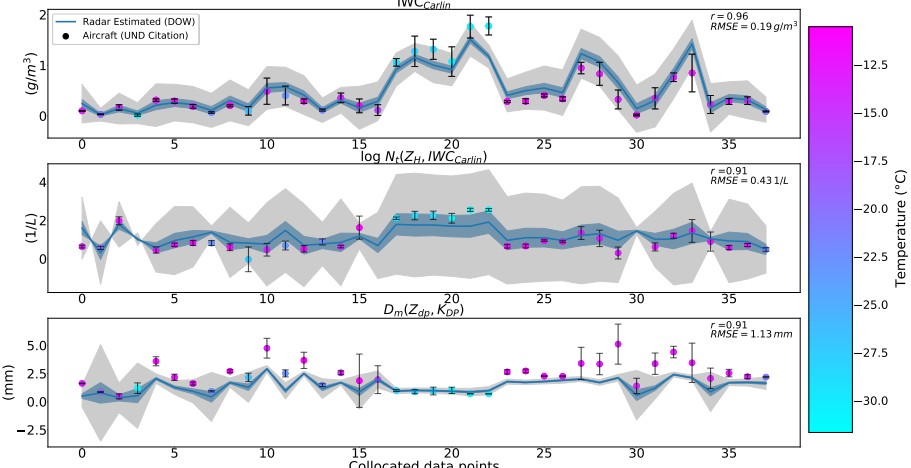

**Figure 5.** Collocated aircraft in situ data in chronological order (colored dots) and the best performing set of ice microphysical retrievals based on RSVP data (solid blue lines) for 10 flight missions. Plots represent from top to bottom IWC$_{\text{Carlin}}$, $N_{\text{t}}(Z_{\text{H}}$,IWC$_{\text{Carlin}})$, and $D_m(Z_{\text{dp}}, K_{\text{DP}})$. Shadings show $\pm 1\sigma$ (gray) and standard error of the mean (blue) calculated via Gaussian error propagation. Colours following the colour bar indicate the respective temperatures in °C. Vertical bars represent in situ standard deviations.

spectral bin model results in a more constrained forecast with respect to the snowfall start time compared to using the IWC($Z_{\text{H}}$,
$K_{\text{DP}}$) retrieval only. The outstanding performance of IWC$_{\text{Carlin}}$ within this accuracy assessment is in line with these findings.

The polarimetric retrieval for $N_{\text{t}}(Z_{\text{H}}$,IWC$_{\text{Carlin}})$ poses a greater challenge (Fig. 5, middle panel). It shows a high variability and more pronounced deviations from the in situ measurements, but still reaches a convincing correlation of r=0.91. Similar to IWC$_{\text{Carlin}}$, $N_{\text{t}}(Z_{\text{H}}$,IWC$_{\text{Carlin}})$ shows an overestimation at warmer and underestimation at colder temperatures. Pronounced $N_{\text{t}}$ deviations at warmer temperatures are visible for the collocated data points 29 and 31, with the former also revealing
large standard deviations on the in situ side. The outlier at data point 9 exhibits $K_{\text{DP}}$ values below 0.1 °km$^{-1}$, which may indicate deficiencies of this retrieval for very low $K_{\text{DP}}$ values, but the in situ values show relatively high variability as well. The $D_m(Z_{\text{dp}}, K_{\text{DP}})$ retrieval (Fig. 5, bottom panel) estimates particle size especially well at colder temperatures. Mixtures of aggregates with different sizes coexisting near the ML (solid region) may explain the high in situ standard deviations at warmer temperatures (e.g., data point 29 where $D_m$ is in excess of 5 mm). Furthermore, the occurrence of an increased num-
ber of aggregates reducing the information content of $K_{\text{DP}}$ can result in a underestimation of $D_m(Z_{\text{dp}}, K_{\text{DP}})$, which makes $D_m$ estimation via polarimetric retrievals close to the freezing level challenging. In line with our analyses, Murphy et al. (2020) indicated larger errors with polarimetric $N_{\text{t}}$ and $D_m$ retrievals in the immediate vicinity of the ML. Despite this underestimation, a convincing correlation of r=0.91 is obtained.

Direct comparisons of both IWC(Z,T) retrievals with the IWC$_{\text{Carlin}}$ retrieval reveals that both non-polarimetric retrievals show
a worse performance in terms of lower correlation and higher RMSE (see Table 3). Note that the use of temperature information from soundings or models would further reduce the performance of IWC(Z,T) retrievals due to increased uncertainties





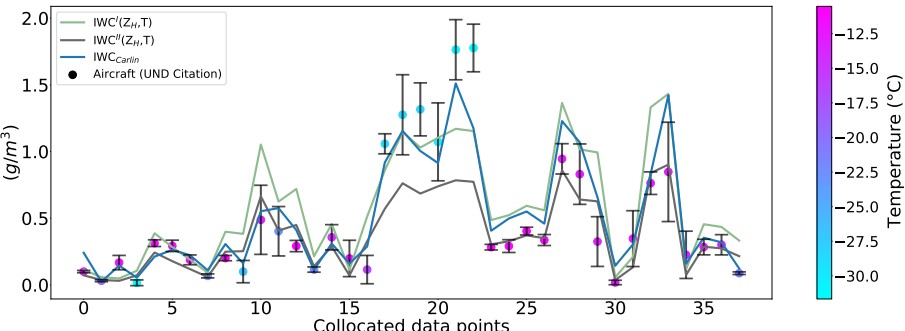

**Figure 6.** Collocated aircraft in situ data in chronological order (colored dots), the IWC$^I$($Z_H$,T) (green), IWC$^{II}$($Z_H$,T) (gray), and IWC$_{Carlin}$ (blue) of RSVP data (solid lines) for 10 flight missions. Vertical bars have the same meaning as in Fig. 5.

compared to in situ temperature data recorded during the flights. Figure 6 indicates that each IWC($Z, T$) shows a better performance in a certain temperature range. Accordingly, we combined them using Eq. 11 for T <= -15°C and Eq. 12 elsewhere. I.e. the DGL located between approximately -10°C to -15°C providing the optimal conditions for depositional growth of ice,

serves here as the boundary for the two retrievals. This combination, hereafter referred to as IWC$_{Comb}$($Z_H, T$), shows promise in estimating IWC with non-polarimetric data more precisely, as it reduces the RMSE to 0.20 and increases the correlation to 0.95 (Table. 3). The bias of the combined method with a value of 0.02 gm$^{-3}$ is slightly closer to 0 compared to IWC$_{Carlin}$. However, a more extensive dataset is needed to corroborate this finding.

Heymsfield et al. (2008) also evaluated non-polarimetric IWC retrieval methods, including the IWC$^I$($Z_H, T$) variant for 95

GHz, using test datasets derived from in situ microphysical measurements. The retrieved-to-measured ratio (RMR) was calculated via dividing the mean or median of the retrieved quantity by the measured quantity, with a selected range of 0.75 < RMR < 1.25 indicating "good" agreement between retrievals and measurements. Their IWC$^I$($Z_H, T$) showed a tendency to underestimate IWC at low temperatures and overestimate it at warm temperatures, consistent with our results. Overall, their non-polarimetric radar-only approach and radar-temperature retrievals yielded a mean (median) RMR of 1.29 (1.20). Our anal-

ysis shows a similar mean RMR of 1.23 for IWC$^I$($Z_H, T$), but a slightly higher median RMR value of 1.59. It is noteworthy that the polarimetric IWC$_{Carlin}$ outperforms IWC$^I$($Z_H, T$) and their radar-only and radar-temperature retrievals, with a mean RMR close to unity (see Table 3).

Figure (7) also demonstrates the pronounced biases associated with the $D_m^I$($Z_h$) and $D_m^{II}$($Z_h$) retrievals, with especially large deviations (strong overestimation) at colder temperatures. This result underlines again the importance of the key variable $K_{DP}$

in ice and reveals the notorious inaccuracies of conventional $D_m$ retrievals. However, these power-laws could potentially be used in combination with polarimetric $D_m$ retrieval in areas near the ML in a hybrid fashion to obtain a more accurate estimate of $D_m$.

Our analysis of $N_t$($Z_H$,IWC$_{Carlin}$) shows good agreement in terms of mean RMR (1.08), whereas none of the $D_m$ retrievals in this evaluation fall within the chosen RMR range, with the best performing $D_m$($Z_{dp}, K_{DP}$) exhibiting a RMR median value



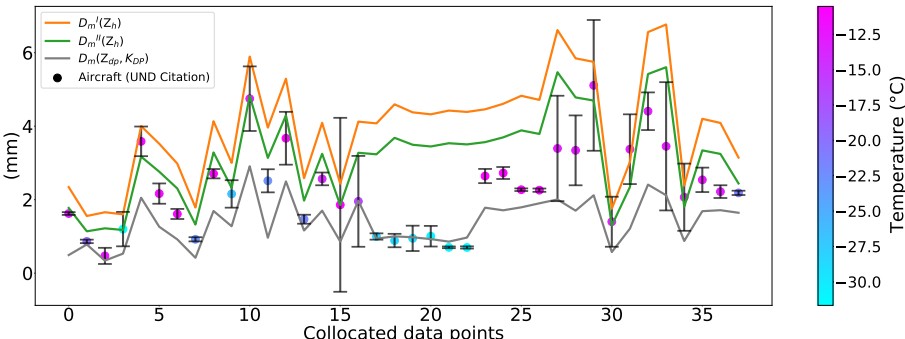

**Figure 7.** Collocated aircraft in situ data in chronological order (colored dots), the $D_m$ power-laws (yellow and green), and the $D_m(Z_{\mathrm{dp}}, K_{\mathrm{DP}})$ retrieval (gray) of RSVP data (solid lines) for 10 flight missions. Vertical bars have the same meaning as in Fig. 5.

close to the lower limit with a value of 0.74. Overall, our analysis is consistent with findings by Murphy et al. (2020), who found a strong underestimation of the $D_m(Z_{\mathrm{dp}}, K_{\mathrm{DP}})$ retrieval near the ML. Similarly, our analysis shows the best results in regions with high $Z_{\mathrm{DR}}$ and $K_{\mathrm{DP}}$, such as in the DGL, and the worst just above the freezing level, where $Z_{\mathrm{DR}}$ and $K_{\mathrm{DP}}$ signatures nearly vanish as a result of aggregation processes (Ryzhkov et al., 1998).

## 6 Discrepancy analyses

For a more detailed analysis and improved understanding, examples of most pronounced discrepancies encountered between in situ and retrieved quantities are spotlighted and presented together with CPI imagery, HVPS samples and/or RSVPs.

On 10 December 2015 at 15:58 UTC (data point 22, see top panel in Fig.5), the aircraft entered the RSVP column at a distance of 15.5 to 20 km from the DOW. In this flight segment, a mean temperature of -31°C was measured by the Citation II at a mean altitude of 5.7 km. The discrepancy between in situ observed IWC and retrieved IWC$_{\mathrm{Carlin}}$ is striking here, with

an underestimation of 0.61 gm$^{-3}$ by the radar-based retrieval. This is the only data point where the IWC discrepancy is as pronounced that even the uncertainty estimates via the standard deviations from both, in situ and retrieval side, do not show any overlap. One possible reason for the underestimation by IWC$_{\mathrm{Carlin}}$ is the temporal mismatch displayed in Fig. 8. Since the aircraft entered at 15:58 UTC and left at 15:59 UTC, the collocation was assigned to the RHI sector scan that began at 15:54 UTC. Thus, it is temporally at the very end of this sector scan. However, the subsequent one might contain signatures that were

already started to be measured at the end of the previous sector scan. For instance, a higher IWC due to advection at the end of the scan could be present. Note that two RHI sector scans are always followed by a PPI scan of 2 min. The following RHI sector scan starting at 16:00 UTC is also close in time to the in situ measurements and shows a higher IWC of 1.39 gm$^{-3}$, reducing the discrepancy to 0.39 gm$^{-3}$. Moreover, choosing this scan leads to overlapping standard deviations, with a retrieval $\sigma$ of 0.45gm$^{-3}$. However, this data point also shows the slight underestimation of IWC$_{\mathrm{Carlin}}$ at colder temperatures.

The most pronounced discrepancy in terms of $D_m$ emerges on 13 December 2015 at 16:17 UTC (data point 29, see lower panel in Fig.5) when the aircraft entered the RSVP column at a distance of 6.5 to 11 km from the DOW. Here, the Citation II



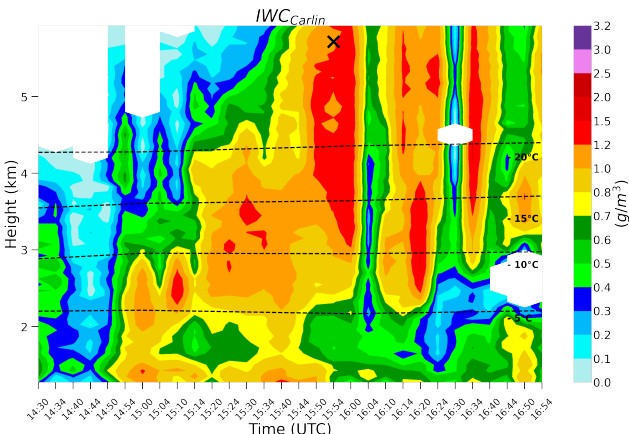

**Figure 8.** RSVPs of IWC$_{\text{Carlin}}$ at ranges between 15.5 km–20 km from the DOW on 10 December 2015 between 14:30 and 16:54 UTC. Overlaid dashed lines display the -5°C, -10°C, -15°C and -20°C isotherms from ERA5 at the DOW location. The collocated data point is indicated by the black cross.

measured a mean temperature of -13°C at a mean altitude of 2.95 km. Retrieved $D_m(Z_{\text{dp}}, K_{\text{DP}})$ strongly underestimates the in situ observed $D_m$ peak exceeding 5 mm, with an underestimation of almost 3 mm. Note that in this case there is no temporal mismatch. Figure 9 illustrates sampled particles captured along the relatively narrow flight path of the aircraft. It seems the

smaller in situ sample is dominated by larger particles, while the larger RSVP volume also accounts for the immediate vicinity where more smaller particles may be present. These possibly affect $D_m(Z_{\text{dp}}, K_{\text{DP}})$ significantly due to averaging and may not properly resolve the regions with enhanced in situ $D_m$ peaks. In addition, the DOW sector scan started at 16:14 UTC and thus the RSVP volume also contains particles not monitored by the aircraft entering the sector later. The particles may be smaller and could explain the discrepancy between in situ observed $D_m$ and retrieved $D_m(Z_{\text{dp}}, K_{\text{DP}})$. It is also worth mentioning that

an in situ $\sigma$ of 1.78 mm is observed along the trajectory, indicating high variability in this region.

    In agreement with the RICE measurements, indicating the presence of SLW with a sudden decrease in frequency (Fig. 10), the CPI example particle images in Fig. 11 also indicate ongoing riming in the flight segment under discussion. Moreover, a clear sagging of the ML in terms of $Z_{\text{DR}}$ and $\rho_{hv}$ (not shown) in the RSVPs supports the riming hypothesis, as since rimed ice particles fall with increased velocities and therefore melt at lower altitudes. Such a sagging signature in radar images

can be associated with riming processes (Kumjian et al., 2016). And since riming, most likely present in this case, results in both higher $\varphi$ and f$_{\text{rim}}$ values than assumed in the derivation of the $D_m(Z_{\text{dp}}, K_{\text{DP}})$ retrieval, the observed underestimation is in line with the known shortcomings of the retrieval for such conditions. Additionally, for riming conditions present the mass-dimension relation assumed for in situ measured IWC, is also not valid anymore. In general, the precise effect of riming on mass-dimension relations is poorly understood (Tridon et al., 2019), but these processes considerably modify inherent

parameters. With increasing degree of riming, higher values of the prefactor $a$ and the exponent $b$ in Eq. (6) are expected. The latter can reach a maximum $b$ close to three (sphere-like geometry) when graupel like particles filled with rime are present.



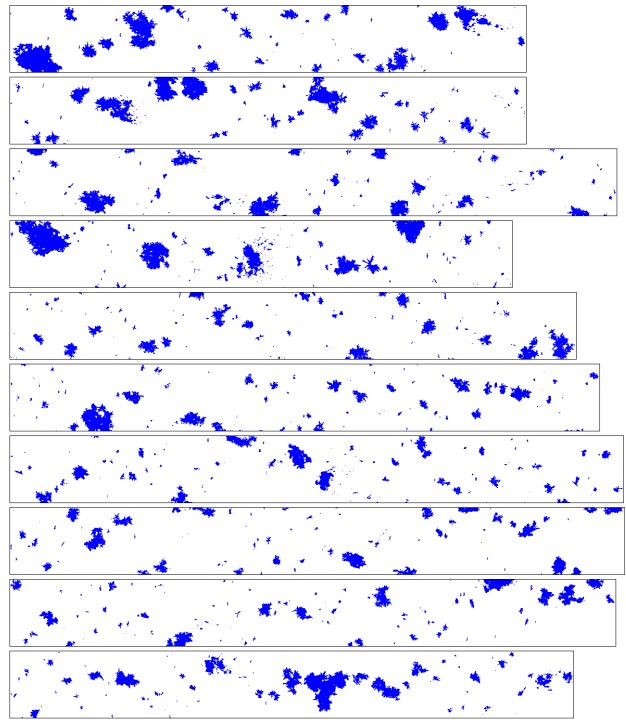

**Figure 9.** HVPS sample images of hydrometeor shadows on 13 December 2015 at 16:18 UTC. The images correspond to the trajectory within the DOW sector scan starting at 16:17 UTC. The height of each panel represents 19.2 mm.

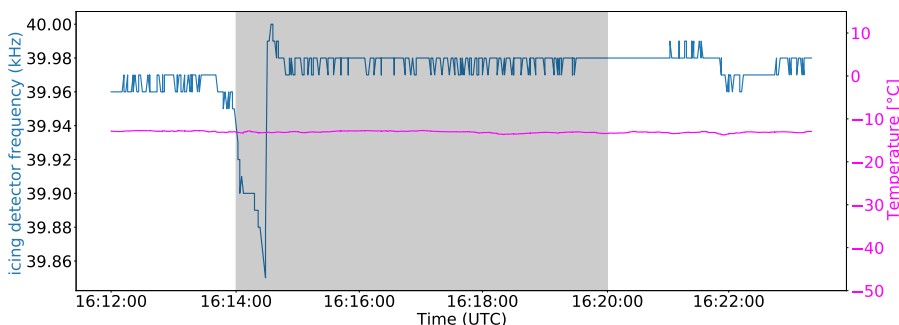

**Figure 10.** RICE oscillation frequency (blue) and temperature at Citation II level (magenta). Icing periods are indicated by a sharp drop in RICE frequency due to ice accumulation at the sensor tip. The shaded area represents the associated flight interval on 13 December 2015.

Thus, the observed overestimation of IWC$_{\text{Carlin}}$ could at least partly be explained by assumed inappropriate parameters in the mass-dimension relationship following Brown and Francis (1995). Using a relation that accounts for more knowledge of ice particle masses (e.g., Heymsfield et al., 2010), that is valid for higher degrees of riming (e.g. Leinonen and Szyrmer, 2015), or that uses multiple mass-dimension relations for different $D_{mm}$ ranges, as proposed in Ding et al. (2020) may reduce deviations



between in situ and retrieved IWC in this case. Only directly measured in situ IWC can provide IWC without the need for any assumptions. However, such measurements were not available during the OLYMPEX campaign (Tridon et al., 2019).

## 7 Conclusions

Data collected during the OLYMPEX campaign (Houze Jr et al., 2017) conducted in late 2015 provided a comprehensive
database including ground-based polarimetric X-band radar measurements and airborne in situ cloud measurements to evaluate radar-based ice microphysical retrievals. In this study, the accuracy of conventional non-polarimetric retrievals is assessed together with a series of state-of-the-art polarimetric retrievals to quantify the benefits of additional polarimetric information and identify the strengths and weaknesses of all of them. RHIs within an azimuthal sector of $22°$ provide vertically high-resolved polarimetric measurements and the RSVP matching methodology introduced adds a moderate degree of averaging in
order to reduce the noisiness of especially phased-based radar measurements like $K_{\mathrm{DP}}$. The matching of the achieved robust radar-based retrievals with the airborne cloud measurements enables the accuracy assessment of retrievals for ice water content IWC, ice particle number concentration $N_{\mathrm{t}}$ and mean volume diameter $D_m$, which are of great value for model evaluation and data assimilation. The key results of the study are as follows:

1. State-of-the-art microphysical retrievals exploiting polarimetric radar measurements to estimate IWC (RMSE = 0.19
415        gm$^{-3}$), $N_{\mathrm{t}}$ (RMSE = 0.43 L$^{-1}$), and $D_m$ (RMSE = 1.13 mm) achieve quite high agreement with airborne in situ measurements, especially at cold temperatures.

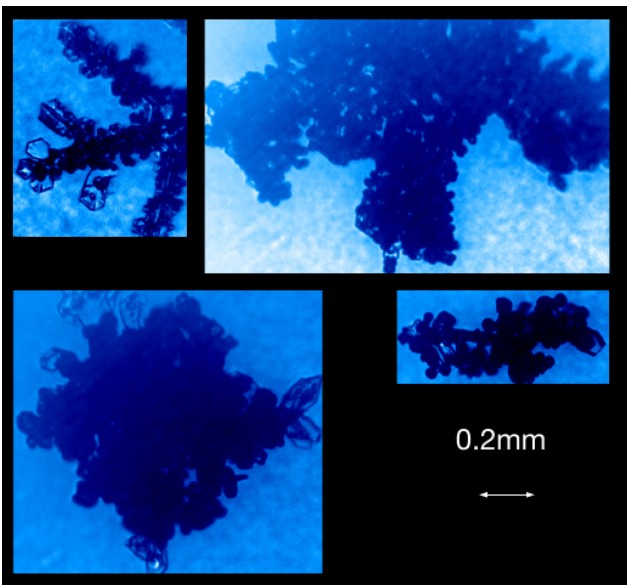

**Figure 11.** Examples of CPI images where rimed crystals were observed. The images were taken on 13 December 2015 between 16:14–16:18 UTC.



2. Overall, polarimetric retrievals are superior to conventional Z-based retrievals, but combinations with non-polarimetric retrievals have potential to improve deficiencies directly above the ML.

3. The hybrid polarimetric IWC$_{\text{Carlin}}$ retrieval outperforms all other IWC estimates in terms of RMSE and shows a high $r$ of 0.96.

4. Compared to IWC retrievals, the $N_{\text{t}}$ and $D_m$ retrievals show larger uncertainties and should be further improved in the future.

This study clearly demonstrates the added value of multiparameter retrievals as proposed by Carlin et al. (2021) for the IWC. Combining the strengths of retrievals can be applied to other ice microphysical properties and ultimately also hints at the potential for developing combinations of polarimetric and non-polarimetric retrievals. Further independent evaluation studies particularly focusing on such hybrid retrievals are required.

Future work using an additional particle classifier (e.g. Praz et al., 2018; Przybylo et al., 2022) capable of identifying and classifying each particle along flight transects with high-resolution image data sampled by particle imagers will provide an even more in-depth evaluation. Specifically, extended classifiers that possibly allow riming degree estimation, as proposed in Przybylo et al. (2022), may provide an avenue for studying riming cases and relating these observations to retrieval assumptions. These classifiers hold the potential to be utilized for the refinement and development of future ice microphysical retrieval methods.

The exploitation of the OLYMPEX data represents another piece of the mosaic towards a comprehensive evaluation of (polarimetric) microphysical retrievals. While previous evaluation studies focused on C- and S-band radar data, this study also emphasizes the added value of X-band radars for the exploitation of microphysical retrievals and related process studies. However, in light of potential applications in model evaluation and data assimilation, C- and S-band radars are of great interest because of their national operational availability. Recently, even more field campaigns have been exploited or their analysis is currently underway. Examples are the Federal Aviation Administration-led In-Cloud ICing and Large-drop Experiment (ICICLE; Bernstein et al., 2021) or the NASA-led Investigation of Microphysics and Precipitation for Atlantic Coast-Threatening Snowstorms (IMPACTS; McMurdie et al., 2019). Two flight legs of a winter storm case from the latter campaign have already been used by Dunnavan et al. (2022) for a radar retrieval evaluation. With respect to previous experimental evaluation studies, the more convincing accuracy of the best performing polarimetric retrievals identified in this study gives us further confidence in their application. The benefit of using a shorter wavelength than of S-band radars, as suggested in Ryzhkov et al. (1998) for the verification of their proposed IWC$^{\text{I}}(Z_{\text{dr}}, K_{\text{DP}})$, indeed exhibited a noticeably lower RMSE in our study (0.24 gm$^{-3}$ vs. 0.4 gm$^{-3}$). Using the same IWC$^{\text{I}}(Z_{\text{dr}}, K_{\text{DP}})$ retrieval as Nguyen et al. (2019), we were further able to achieve half the RMSE and a systematically higher correlation with our method based on RHI-sector scans, even though both studies used an X-band radar. More recent retrievals were able to achieve even smaller RMSEs. The almost consistent underestimation of $D_m(Z_{\text{dp}}, K_{\text{DP}})$ shown in Murphy et al. (2020) is in line with our results. However, in contrast to their study, we could not only attribute the deficits to warmer temperature regimes, but also demonstrated the accurate estimation at cold temperatures. As a



result, an important open question for future research concerns the deficient performance of retrievals directly above the ML and requires new approaches to obtain accurate estimators in this region.

*Data availability.* OLYMPEX data used in this study were obtained from the NASA EOSDIS GHRC OLYMPEX data archive (doi: http://dx.doi.org/10.5067/GPMGV/OLYMPEX/DATA101 and doi: http://dx.doi.org/10.5067/GPMGV/OLYMPEX/DOW/DATA201; last access: 11 November 2022). The ERA5 data are stored at the Climate Data Store from the ECMWF and are accessible via https://cds.climate.
copernicus.eu (last access: 11 November 2022).

*Author contributions.* AB and ST jointly developed the concept and methodology for this work. Data handling and analysis was performed by AB with contributions from MM. Visualization was carried out by AB, who also led the writing with input from ST. AJH processed the Citation II data collected during OLYMPEX and provided scientific expertise on in situ data. All authors contributed to the proof-reading and added valuable suggestions to the final draft.

*Competing interests.* The authors declare that they have no conflict of interest.

*Acknowledgements.* The research was carried out in the framework of the Priority Programme SPP-2115 "Polarimetric Radar Observations meet Atmospheric Modelling (PROM)" funded by the German Research Foundation (DFG). We also thank all the participants of OLYMPEX for collecting the data used in this study. Similarly, we acknowledge the support of Kai Mühlbauer and the open-source radar library wradlib (https://docs.wradlib.org/en/stable/index.html, last access: 11 November 2022) regarding the processing of radar data.

*Financial support.* This research has been supported by the Deutsche Forschungsgemeinschaft (grant nos. TR 1023/13-1 and VO 1504/5-1). Andrew J. Heymsfield acknowledges the support of NASA under grant 80NSSC20K0897.



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
