# Peer review of "Evaluation of polarimetric ice microphysical retrievals with OLYMPEX campaign data"

_EGUsphere, 2022_

## Referee Comment (RC2)

Review of egusphere-2022-1488

The manuscript "Evaluation of polarimetric ice microphysical retrievals with OLYMPEX campaign data" is satisfactorily written, and the content is within the journal's scope. The authors introduce a new way of radar data processing, RSVP. They compare polarimetric and standard X-band radar retrievals with in situ aircraft measurements of IWC, $D_m$, and $N_t$, suggesting that dual-pol information improves the microphysical retrievals. Minor revision is the recommendation.

Specific comments:

Lines 140-145: Brown and Francis (1995) (BF) relation derived for cirrus clouds may not represent IWC properly in different cloud types and environmental temperatures. Authors may want to comment a bit more on the BF usage (there is only a short comment in section 5 beginning about the uncertainty of the assumed mass-dimension relation – line 285).

Line 167: Reword the part about the "confusion among the experts"; it is probably a typo in Murphy et al. (2020) – their multiplier is reciprocal to multiplier in (8).

Lines 224-229: The factor $\alpha_0$ does not vary with the degree of riming $f_{rim}$; it is constant. The authors probably wanted to emphasize that the prefactor $\alpha$ in the snow density relation, $\rho_s = \alpha D^{-1} = \alpha_0 f_{rim} D^{-1}$, varies with the degree of riming.

Line 231: There is no reference to Table 2 in the body text.

Line 311: Provide a reference for the temperatures - the text is difficult to follow this way.

Line 318: The same comment as for Line 311.

Line 320: Did the authors mean Kdp<0.01 deg/km here?

Line 388: Remove "as" from the sentence.

---

## Author Comment (AC1)

Review of "Evaluation of polarimetric ice microphysical retrievals with OLYMPEX campaign data", by Armin Blanke, Andrew J. Heymsfield, Manuel Moser, and Silke Trömel, egusphere-2022-1488.

**Response to reviewer 1**

Dear reviewer,
We are very grateful for your valuable feedback and suggestions which helped us to improve the manuscript. The manuscript has been thoroughly revised and point-by-point responses have been prepared. Please find below our replies, highlighted in blue, along with your suggestions. The revised manuscript is also provided with tracked-changes for clarity.

Specific comments:

Line 100-104, is rhohv > 0.7 the only quality control used here for Kdp calculation? Is SNR also been taken into account?

For the calculation of Kdp, we already use the quality assessed filtered PHIDP_F from the OLYMPEX dataset. In addition, only data within a maximum range of 20 km and at maximum altitudes of 7 km (mean 4 km) of the research aircraft are used for our accuracy assessments. The lowest SNR included in our analyses was 9.35 dB, and the mean SNR of all collocated data points used was 25.47 dB.

Line 113-115, regarding 2D-S and HVPS probes, did the authors checked for the down times (mostly due to system overload)? Any dual polarimetric signal patterns with the down times of the probes data? And what is the fraction of the down times to the useful time steps during the flight?

The data processing of the 2D-S and HVPS includes dead-time correction. SPEC probes, which are used here, handle dead time in such a way that the data can be accurately corrected. Gurganus and Lawson (2018, doi: 10.1175/JTECH-D-17-0202.1) also determined the dead time for SPEC probes to be very small. Overall, our in situ derived microphysical properties are mostly based on HVPS data (particularly $D_m$ and IWC), which rarely go into overload.
Thus, the very good performance of the 2D-S and HVPS and the dead-time correction result in a neglectable error due to dead times (fractions of seconds) of the in situ instruments. In addition, shattered particles are rejected using corrections for shattering, now mentioned in Lines 117 - 119: 'Standard processing and correction options with SODA (Software for OAP Data Analysis, provided by A. Bansemer, National Center for Atmospheric Research/University Corporation for Atmospheric Research UCAR, 2013) were applied including shattering and dead time corrections.'

Line 245-250, for the RSVP, when RHIs are averaged, since each RHI has its own averaged time steps, is the time difference also interpolated? Also, for gates to gates average, the distance for each gate to the center of vertical profile is different, did the authors used any technique like distance inverse weighting to take this into consideration?

Since all 22 RHIs are considered for averaging, the resulting averaged RHI is representative of the entire time span of the sector's scan duration, starting with the first RHI and ending with the last. In this case, no time differences need to be interpolated. In addition, techniques such as inverse distance weighting were not used, because we are comparing collocated radar volumes with flight

trajectories. We do not compare the center (or any other specific location within the sector) to a point measurement of the aircraft. Due to the duration of the consecutive RHI scans and the changing trajectories for each case, it is hard to estimate whether weighting by the closest RHI in time or space makes more sense. Instead, the average of both, the RHIs and the trajectories are not weighted.

---

## Author Comment (AC2)

Review of "Evaluation of polarimetric ice microphysical retrievals with OLYMPEX campaign data", by Armin Blanke, Andrew J. Heymsfield, Manuel Moser, and Silke Trömel, egusphere-2022-1488.

**Response to reviewer 2**

Dear reviewer,
We are very grateful for your valuable feedback and suggestions to improve the manuscript. The manuscript has been thoroughly revised and point-by-point responses have been prepared. Please find below our replies highlighted in blue along with your suggestions. The revised manuscript is also provided with tracked-changes for clarity.

General comments:

The manuscript "Evaluation of polarimetric ice microphysical retrievals with OLYMPEX campaign data" is satisfactorily written, and the content is within the journal's scope. The authors introduce a new way of radar data processing, RSVP. They compare polarimetric and standard X-band radar retrievals with in situ aircraft measurements of IWC, Dm, and Nt, suggesting that dual-pol information improves the microphysical retrievals. Minor revision is the recommendation.

Specific comments:

Lines 140-145: Brown and Francis (1995) (BF) relation derived for cirrus clouds may not represent IWC properly in different cloud types and environmental temperatures. Authors may want to comment a bit more on the BF usage (there is only a short comment in section 5 beginning about the uncertainty of the assumed mass-dimension relation – line 285).

Thank you for this suggestion. An extended explanation on the usage of BF is given in Lines 148-153:
'Even though parameters a and b of the mass-dimension relationship vary with the environmental conditions and particle shapes (Baker and Lawson, 2006), constant standard parameters are used in this study which reasonably represent the mean ice water content, especially for ice crystal aggregates. Tridon et al. (2019) confirmed that the aggregates observed during OLYMPEX can mostly be described by a quite narrow range of mass-size relations. In single situations with large aggregates or intense riming processes, however, the fixed parametrization may underestimate the ice water content (see also Heymsfield et al., 2023).' We included Baker and Lawson (2006; doi: 10.1175/jam2398.1) and Heymsfield et al. (2023 ; doi: 10.1175/JAMC-D-22-0057.1) as references and also referred to Moser et al. (2023; doi: 10.5194/acp-2023-44) with respect to the operating principles, uncertainties and limitations of the 2D-S and HVPS. Please see Lines 111 - 112.

Line 167: Reword the part about the "confusion among the experts"; it is probably a typo in Murphy et al. (2020) – their multiplier is reciprocal to multiplier in (8).

Yes, it is likely a typo in Murphy et al. (2020). We rephrased as follows: ' This equation differs from Eq. (5) in Murphy et al. (2020) showing, due to a typo, the reciprocal multiplier when converting $D_{mm}$ to $D_m$.' Please see Lines 175 - 176.

Lines 224-229: The factor $\alpha_0$ does not vary with the degree of riming frim; it is constant. The authors probably wanted to emphasize that the prefactor $\alpha$ in the snow density relation, $\rho s = \alpha D^{-1} = \alpha_0 f_{rim} D^{-1}$, varies with the degree of riming.

Indeed, thanks for pointing it out. We clarified Eq. (20): '… where $\rho_s$ is expressed in g cm$^{-3}$, $\alpha_0$ is a constant, and the prefactor $\alpha_p$ varies with the degree of riming $f_{rim}$, which ranges from 1 for unrimed ice to 5 for heavily rimed ice.' Please see Lines 235-237.

Line 231: There is no reference to Table 2 in the body text.

Thanks, we are now referring to Table 2 in Lines 242 - 243.

Line 311: Provide a reference for the temperatures - the text is difficult to follow this way.

Thanks for raising this point. With colder temperatures we refer to the range $T \lessapprox -27°C$ and with warmer temperatures we refer to $T \gtrapprox -14°C$. We included this information in brackets. Please see Lines 320 - 321.

Line 318: The same comment as for Line 311.

We indicated again the according temperature ranges. Please see Lines 328 – 329.

Line 320: Did the authors mean Kdp<0.01 deg/km here?

No, we are not referring here to the filtering criterion mentioned in Line 274, in fact we mean that the data point exhibit a low Kdp value below 0.1 deg/km (exact value: 0.08 deg/km).

Line 388: Remove "as" from the sentence.

Thanks, removed in Line 398.